# Reproducibility and Stability Analysis in Metric-Based Few-Shot Learning

## Abstract

We propose a study of the stability of several few-shot learning algorithms subject to variations in the hyper-parameters and optimization schemes while controlling the random seed. We propose a methodology for testing for statistical differences in model performances under several replications. To study this specific design, we attempt to reproduce results from three prominent papers: Matching Nets, Prototypical Networks, and TADAM. We analyze on the miniImagenet dataset on the standard classification task in the 5-ways, 5-shots learning setting at test time. We find that the selected implementations exhibit stability across random seed, and repeats.

## 1 Introduction

Concern about reproducible science has grown in the machine learning field and the scientific community as a whole in the past decade. It has been found that a significant proportion of published research could not be reproduced across multiple scientific disciplines including cancer biology [8], psychology [4] and machine learning [17, 25].

In this article, we offer a principled methodology for evaluating replication of machine learning experiments, and apply it to prominent algorithms in the domain of few-shot learning. Notably, few-shot learning is a key area for applications of machine learning in widespread and data-poor environments [13]. Low data availability and quality arguably puts results at an even higher risk of uncontrollable variability. Therefore, it is imperative that we have sound statistical tools and methods for machine learning algorithm evaluation.

Our contribution is three-fold:

- We highlight major challenges faced when trying to replicate results from three metric-based few-shots learning articles.
- We establish a procedure to evaluate few-shot learning algorithms through a specific design of experiment and a series of tests.
- We explore and compare the performances of 3 algorithms in depth and present results on the miniImagenet task.

### 1.1 Literature review

**Terminology** We adopt a terminology for reproducible research that distinguishes the following terms: repeat, replicate, reproduce [10, 5]. A *repeat* indicates an attempt to achieve the same results by the same lab with the same setup. A *replication* is a repeat performed by researchers at an independent lab. Finally, a *reproduction* is the attempt to achieve the same results with some differences to the setup and at an independent lab.

Submitted to the 2019 International Conference on Learning Representations Reproducibility Workshop

**Model complexity** In the past few years, the complexity of machine learning models has grown dramatically [18, 19]. The proliferation of deep neural network models has been accompanied by a general increase in model configuration complexity. Model capacity and topology, hyper-parameters, regularizers, and optimization regimes all contribute to model performance. Deep learning is typically performed with hardware accelerators (e.g. GPU, TPU). Their non-deterministic runtime behaviour further complicates even a simple re-run of experiments.

It has been demonstrated in language modeling that hyper-parameter selection is a significant factor in model performance, and can in fact dominate architectural differences [25]. In the Reinforcement Learning setting, current state-of-the-art algorithms are sensitive enough to random seed alone that the top-N means often reported are not representative of true performance [17]. These trends have informed contemporary ML scholarship and will continue to influence the way applied research is performed.

**Producing reproducible research** Recommendations in the literature are:

- Control and measure factors of variation: hyper-parameters, regularization, random seed, optimization regime [17, 25, 23]
- Significance testing and error analysis [31, 17, 25]
- Ablation studies [23]
- Release of hyper-parameters and the method by which they were selected [10, 17, 9]
- Workflows and processes for consuming results from other labs as well as initial data capture required for reproduction [10, 9, 35]

## 1.2 Motivation and scope

**Motivation** The machine learning community mostly use deductive reasoning over inductive reasoning for applied research. In both cases, researchers want to build a generalizable model of a phenomenon and study it by collecting and then analyzing measures of a specific metric on specific tasks.

The scientific advances (theory, methodology, results), and produced artifacts (data, code) become tools for the community to use. Everyone can use the tools to create new research and produce new scientific advances, and use the research in applications impacting the world locally or globally. On top of that, being able to criticize or invalidate theories and models [12] or prove the proposed theory, methodology or results wrong (skepticism, [20]) is highly important.

We argue that studying reproducibility of the results in this context can help the applied research community produce better research tools and develop better practices.

**Scope** In this context, we have selected three few-shot learning articles using metric-based learning to study and compare their reported results on a specific task and a specific dataset. We use the official publicly available implementations provided by the authors when available, and the community endorsed one otherwise. We document the process to reproduce the reported results and analyze the results we were able to obtain by changing the hyper-parameters settings if need be.

We investigate how the selected scientific contributions exhibit different behaviors subject to different random seeds repeats and hyper-parameter changes. We also define an example of design of experiment when using repeats and fixed random seeds. We test different settings of hyper-parameters, seed the different pseudo random number generators and repeat the training for every hyper-parameter configuration multiple times. This specific setup allows us to use linear mixed models to reason about our research hypothesis while taking into account the reproducibility concerns.

## 2 Methodology

### 2.1 Experimental protocol

**Dataset** We use the miniImagenet dataset proposed by Vinyals et al. [36] to perform our experiments. There are 100 classes divided into 3 splits comprised of 64, 16, and 20 classes for meta-train, meta-validation and for meta-test, respectively. Each class has 600 samples of 84 x 84 images. To construct

the tasks, we sample 5 classes uniformly and 5 training samples per class uniformly. We use the (meta-) train, validation and test splits from Ravi and Larochelle [30].

**Models** For this review we have selected three metric-based few-shot learning models: Matching Networks [36], Prototypical Networks [34], and TADAM [27]. These models represent the state of the art in the 5-shot case for 2016, 2017, and 2018, respectively.

**Replicability effort** We identified the official or community-endorsed implementation for each model. We then aimed at replicating results using the default hyper-parameters and setting random seeds (see 5.1). When part of the implementation or some hyper-parameters values were missing, we complemented the existent to the best of our knowledge and ability, thus engaging in a reproducibility effort.

**Variability analysis** We ran multiple experiments using hyper-parameter search and recorded model test accuracy in order to perform a study of the variability of state-of-the-art few-shot learning models.

## 2.2 Description of intended analysis

The first step of the procedure is to propose a set of research hypotheses. For our study we select 3 research hypothesis that we want to test:

- H1: The results across runs of the same algorithm using the same configuration and the same random seed are stable

- H2: The results across runs of the same algorithm using the same configuration but a different random seed are stable

- H3: In our group of experiment rerunning the same experiment using the same hyper-parameters configurations and same random seed yield stable results

The second step is to propose a design of experiment to generate the data and test our hypothesis. In order to do so, we perform seeded repeated runs of randomly sampled hyper-parameters configurations for a given implementation of an algorithm. We give more information about the sampling scheme of the configuration in Section 4.

The third step of the procedure is to use statistical tools to falsify or not our set of hypothesis. To do so we define a statistical model suited to analyze clustered data and give more detail in the following section.

## 2.3 Statistical tests

To measure the difference of means between groups in the presence of noisy clustered observations, one can use linear mixed models [14], [7] or hierarchical Bayesian models [15]. Since we control almost completely the environment where the experiments are run, we can define a specific design to reason about statistical reproducibility while comparing the results of different runs of different algorithms. For each sample, we retrieve the information about the experiment name, the hyper-parameter configuration, the random seed used, the repeat identifier and the test accuracy on the meta test split.

In our setup we have of $N \times D$ features $\mathbf{X}$ corresponding to a contrast matrix in our case (one-hot encoded experiment vector) and $N$ measures of the metric $\mathbf{y}$. We can estimate the effects of each experiment with the linear regression model:

$$\mathbf{y} = \mathbf{X}\boldsymbol{\beta} + \alpha + \epsilon,$$

where $\boldsymbol{\beta} \in \mathbb{R}^D$ is the slope vector, $\alpha \in \mathbb{R}$ is the intercept, and $\epsilon \sim \text{Normal}(\mathbf{0}, \mathbf{I})$ is random noise. In our setup, $\boldsymbol{\beta}$ and $\alpha$ are "fixed effects": we want to measure the difference between groups with constant effects across our dataset $(x, y)$. To achieve this, we maximize the likelihood $\mathbf{y} \sim \text{Normal}(\mathbf{X}\boldsymbol{\beta} + \alpha, \mathbf{I})$ to find point estimates of $\boldsymbol{\beta}$ and $\alpha$ that fit the data. With our design, we know that there is a structure in the data generating process and that the observations $(x, y)$ are not i.i.d. To circumvent this modeling problem we can rewrite our linear model:

$$\mathbf{b} \sim \text{Normal}(\mathbf{0}, \sigma^2 \mathbf{I}) \tag{1}$$

$$\mathbf{y} = \mathbf{X}\boldsymbol{\beta} + \mathbf{Z}\mathbf{b} + \alpha + \epsilon. \tag{2}$$

Where $\boldsymbol{\beta} \in \mathbb{R}^P$ is our slope vector, $\alpha \in \mathbb{R}$ is the intercept, and $\epsilon \sim \text{Normal}(\mathbf{0}, \mathbf{I})$ is the random noise vector. To model the clusters, we introduce $\mathbf{Z}\mathbf{b}$, where $\mathbf{Z}$ is the $n \times q$ model matrix for the $q$-dimensional vector-valued random-effects variable, $\mathbf{B}$, whose value we are fixing at $\mathbf{b}$. $\mathbf{b}$ is normally distributed with variance component parameter $\sigma^2$. In this setting we can rewrite the *conditional* distribution of $\mathbf{y}$ given $\mathbf{B} = \mathbf{b}$ such as $(\mathbf{y}|\mathbf{B} = \mathbf{b}) \sim \text{Normal}(\mathbf{X}\boldsymbol{\beta} + \alpha + \mathbf{Z}\mathbf{b}, \sigma^2 \mathbf{W^{-1}})$.

The $\mathbf{b}$ are "random effects" that vary across the population. Because of equation 1, we have $\mathbb{E}[\mathbf{b}] = 0$, and the dependent variable mean is captured by $\mathbf{X}\boldsymbol{\beta} + \alpha$ when we marginalize over all the samples. The random effects component $\mathbf{Z}\mathbf{b}$ captures variations in the data, it can be interpreted as an individual deviation from the group-level fixed effect.

In our context, we can write the model as follows:

$$\mathbf{metric_{ijk}} = (A + \alpha_{0j} + \alpha_{1k}) + \boldsymbol{\beta} Experiment_i + \epsilon_i \tag{3}$$

$$\mathbf{metric_{ijk}} = A + \boldsymbol{\beta} Experiment_i + (\alpha_{0j} + \alpha_{1k} + \epsilon_i) \tag{4}$$

$$\mathbf{metric_{ijk}} = A + \boldsymbol{\beta} Experiment_i + \varepsilon_{ijk} \tag{5}$$

Where $A$ is the intercept, $\boldsymbol{\beta}$ is a vector of parameters and $Experiment_i$ is a one hot vector of experiments for the observation $i$. We can regroup all the random effects, where $alpha_{0j}$ is a random effect associated with an observation from a random seed $j$, and $alpha_{1k}$ is associated to an observation from a repeat $k$. Finally, it is possible to regroup all the nuisance parameters in $\varepsilon_{ijk} = (\alpha_{0j} + \alpha_{1k} + \epsilon_i)$.

## 2.4 Estimation of the random and fixed effects

To estimate the parameters of the linear mixed model defined in equation 5 we use an implementation in R with the lme4 package [6]. The estimates for the random effects and the fixed effects estimated with lme4 can be augmented with the lmerTest package [22] to add corrected degrees of freedom for the p-values [21], [32].

# 3 Experiments

## 3.1 Matching networks

Matching networks [36] is one of the first metric-based few-shot learning algorithms.

**Official implementation** There is no official implementation provided by the authors. We used the reproduced implementation from a later article [30], which attains results comparable to the original article and is cited heavily by newer papers. This implementation [29] is in torch7.

**Replicability effort and challenges** The technical specifications provided with this implementation were not compatible with our more recent hardware (Tesla P100, Tesla V100). We identified through experimentation the versions of Ubuntu, Cuda, and torch dependencies that worked together for our settings. We changed the C++ compiler flags in the torch IPC dependency to be compatible with those versions.

We were able to replicate the results from [30] with the default parameters without full conditional embedding (FCE), but not the original article results. This is potentially due to different train/validation/test splits between those two articles. All further experiments have been run with FCE enabled, as this proved to be an improvement over basic embeddings according to both [36]'s and [30]'s authors. For more details, see Appendix A.1.

## 3.2 Prototypical networks

Prototypical Networks [34] consist of a convolutional neural network learning a non-linear mapping of the input into an embedding space, in which a nearest neighbor classification is performed by computing distances to prototype representations. We focus on their miniImagenet 5-shot results.

**Official implementation** The authors released their code in an official GitHub repository [33], without parameters and data loading functions for miniImagenet. This shifted our effort to replicate to an intent to reproduce, as part of the implementation was missing.

**Reproducibility effort and challenges** We wrote a dataloader for miniImagenet, as the code released by the authors did not support this dataset on which they however report results. Some training hyper-parameters were not specified in the article ; we found the missing values in other repositories reproducing the results [11, 26] and open issues discussions [24]. We were unable to reproduce the results from the article, obtaining 59.01% ($\pm$ 0.73) accuracy at best in lieu of the expected 68.20% ($\pm$ 0.66) when running the default configuration.

In an effort to improve on these results, we normalized the input over miniImagenet and varied multiple hyper-parameters values. With these settings, we obtained a top accuracy of 62.50% ($\pm$ 0.53) (see Table 6). For more details, see Appendix A.2.

## 3.3 TADAM

TADAM [27], is the method of metric scaling and metric task conditioning that extends the original Prototypical Networks algorithm. Additionally they analyze the impact of varying feature extraction topology and the parameters defining the optimization scheme. The final architecture uses a ResNet-12 feature extractor [16] and a FiLM-ed multilayer task encoder [28]. Finally, they find that co-training the feature extractor on a supervised task improves generalization.

**Official implementation** The implementation made publicly available by the authors [26] worked with the tensorflow-gpu (version 1.13.1) Docker image from Dockerhub [3]. The provided data loading code and hyper-parameters were sufficient to replicate the results reported in the paper on miniImagenet.

**Replicability efforts** We used the default set of hyper-parameters and were able to reproduce the results presented in table 1 in TADAM [27].

**Challenges** The co-training strategy makes controlling the random seed more difficult than naively setting the runtime and library random seeds. The co-training implementation involves different Tensorflow managed sessions creating different graphs multiple times during the training is funciton-ally incompatible with the Adam optimizer. Although the TADAM implementation permits usage of Adam, we were not able to easily modify the training regime in such a way that would make the algorithm train properly.

## 3.4 TADAM Prototypical

In addition to the Prototypical Networks experiment above, we attempt to repeat results from a baseline implementation for the TADAM research project [27] wherein the authors successfully reproduced the original reported results.

**Official implementation**

The official implementation is available on GitHub as part of TADAM and can be enabled by specifying configuration flags and hyper-parameters corresponding to Prototypical Networks. The implementation is in Tensorflow and differs quite a bit from the original in PyTorch.

**Repeatability efforts**

We attempt to repeat the original experiment and achieve the same results as reported for the baseline of prototypical networks in TADAM [27].

The correct hyper-parameters and flags used to define the runtime behaviour corresponding to Prototypical Networks were found after contacting the original authors, and by comparing the original paper and implementation to the TADAM implementation.

We achieved 68.09% ($\pm$ 0.23) test accuracy vs the original reported 68.9 ($\pm$ 0.3).

Table 1: Random effects ANOVA

|  | npar | logLik | AIC | LRT | Df | Pr(>Chisq) |
|---|---|---|---|---|---|---|
|  | 15 | 1813.832 | -3597.664 | NA | NA | NA |
| (1 \| seeds) | 14 | 1813.832 | -3599.664 | 0 | 1 | 1 |
| (1 \| repeats) | 14 | 1813.832 | -3599.664 | 0 | 1 | 1 |

Table 2: Means comparisons

|  | Estimate | Std. Error | lower | upper | Pr(>\|t\|) |
|---|---|---|---|---|---|
| **m-net-adam** | **-0.010969** | 0.005554 | -0.021866759 | -0.0000707299 | 4.853417e-02 |
| m-net-sgd | 0.006967 | 0.005423 | -0.003674030 | 0.0176081909 | 1.991716e-01 |
| protonet-adam | 0.00278 | 0.009015 | -0.0149106977 | 0.020466783 | 7.580161e-01 |
| protonet-sgd | -0.001042 | 0.008971 | -0.0186452120 | 0.016560948 | 9.075433e-01 |
| **tadam-adam** | **-0.020189** | 0.008699 | -0.0372574009 | -0.003119772 | 2.048393e-02 |
| tadam-sgd | -0.000014 | 0.005410 | -0.0106336415 | 0.010605801 | 9.979483e-01 |

## 4 Analysis of experiments

We use the hyper-parameters configurations reported in table 3 to launch our seeded repeated random searches. We also seed the random search to be able to repeat the same experiment twice with the exact same hyper-parameters configuration. We sample 5 hyper-parameters configurations, for each configuration we use 5 different random seeds, we repeat the training 5 times for each random seed. This amounts to 125 configurations per experiment. Overall, we run 12 experiments for a total of 1074 configurations.

From those configurations, we fit the linear mixed model defined in equation 5. Our goal is to quantify the variability in the error linked to the seeds $alpha_{0j}$, quantify the variability in the error linked to the repeat $alpha_{1k}$ and estimate the differences in performances between the different experiments and algorithms.

We first perform likelihood ratio tests for each random effect added to the model, this confirm that adding any of them doesn't significantly change the likelihood in table 1. We can't reject H1 and H2 and confirm that the implementations performances do not vary significantly for the conditions we defined. To verify H3 we first need to test if a difference exists between all the experiments. We can use an ANOVA with a correction for the degrees of freedom for the number of comparisons performed [21]. Table 4 confirms that there is significant difference in the experiments accuracy means. The last part of our analysis compares the means of reruns of the same experiments. In table 2 we compute the means difference and provide standard errors and a 95% confidence interval of our estimators. Among all the comparisons, only 2 are statistically significant. Indeed, the difference between the 2 different runs of Matching Networks with the adam optimizer and TADAM with the adam optimizer have statistically significant means. In that sense we reject H3 and confirm that rerunning the same experiment using the same hyper-parameters configurations and same random seed can yield non-stable results.

## 5 Notes on randomness

### 5.1 Determinism

To make the behavior of each model as deterministic as possible, we set the random seeds for every library used in the implementation. For Prototypical Networks, this includes seeding the `random`, `numpy`, `torch` and `torch.cuda` python modules, as well as the `PYTHONHASSEED` environment variable ; for Matching Networks the `torch`, `math`, `cutorch` LUA modules ; for TADAM and the TADAM Prototypical Networks implementation the `random_ops` TensorFlow module and the `numpy` python module.

As a control, we fixed the value of the seed to 7654 and ran 10 times the same experiment with default parameters. Only the Prototypical Networks implementation has a fully deterministic behavior (see Table 5).

Table 3: Experiments hyper-parameter search space for Adam and SGD optimizers

| Algorithms | TADAM | Proto nets | Matching nets |
|---|---|---|---|
| **Learning rate** | $\mathcal{U}(0.1, 0.02)$ | $\mathcal{N}(0.005, 0.0012)$ | $\log\mathcal{U}(0.0001, 0.1)$ |
| **LR decay rate** | $\mathcal{N}(10, 1)$ | 0.5 | $\log\mathcal{U}(0.00001, 0.01)$ |
| **LR decay period (batch)** | 2500 | $\mathcal{U}(500, 2000)$ | 1 |
| **Query shots per class** | $\mathcal{U}\{16, 64\}$ | 15 | $\mathcal{U}\{5, 30\}$ |
| **Pre-train batch size** | $\mathcal{U}\{32, 64\}$ | - | - |
| **N-Way** | 5 | 5 | 5 |
| **N-Shot / support set** | 5 | 5 | 5 |
| **Number of tasks per batch** | 2 | 1 | 1 |
| **Batch size** | 100 | 100 | 500 |
| **Early stop (epochs)** | | - | 20 |
| **Training steps (batches)** | 21K | 10K | 75K |
| **Test episodes** | 500 | 600 | 600 |

Table 4: Linear Mixed Model fixed effects results

| | Sum Sq | Mean Sq | NumDF | DenDF | F value | Pr(>F) |
|---|---|---|---|---|---|---|
| **experiments** | 12.94413 | 1.176739 | 11 | 18.92112 | 642.001 | 6.497355e-22 |

## 5.2 Other sources of randomness

There are additional sources of variability between different implementations, which need to be addressed to perform a proper comparison. They do not, however, affect different runs with different seeds for a given implementation, except for inherent differences due to parallelism on CPU and GPU.

Some implementations:

- call a random number generator at an execution point placed before the episodes data generation, hence changing the state of the random number generator,

- generate the episodes data in advance, others generate it for each episode on the fly: different states of random number generator are involved in the data generation process,

- start training at different states of the random number generator: random number sequences for training algorithms differ,

- use various languages, various libraries and different versions of those, which can have different algorithms for random number generation.

## 6 Discussion

Fostering reproducible research is not as easy as putting code on GitHub. There are often undocumented sources of variation be it dataloaders, hyper-parameters, or proper description and availability of dependencies.

The trend of large-scale compute-intensive ML experiments has caused concern in the community about the ability of smaller and/or non-industrial labs to replicate. Inability to exactly re-run an experiment does not preclude reproducibility and should not discourage research in the field. In some high-energy physics experiments, there are fundamental limitations to independent experimental setup on instruments [10]. As machine learning practitioners, we are well-equipped to build tools to automate or streamline the important process of proper experiment management.

Table 5: Control analysis accuracy for a set seed.

| - | Matching Nets | Proto-networks | TADAM | Proto nets* |
|---|---|---|---|---|
| **Accuracy** | 50.76 ($\pm$ 0.54) | 56.23 ($\pm$ 0.00) | 71.96 ($\pm$ 2.06) | 66.68 ($\pm$ 0.18) |

The current trend of releasing code with papers is a fantastic step in the right direction for the community. International conferences started encouraging reproducibility in the recent years [2]. The Papers with Code project [1] is a good way to organize and track these code releases. It can provide visibility for papers with high quality reliable implementations, and therefore offers an incentive to the community.

We encourage researchers to publish not just the core implementation of their papers, but also descriptions of their actual operational environments including OS version, libraries version, GPU/TPU models, etc. Linux containers and tooling like Docker with Dockerfile or Singularity make this an achievable goal for many research teams. Additionally it is important to document and report the method by which hyper-parameters were selected and performance significance testing to analyze their effects [17]. Release of this data can allow other researchers to replicate analysis of algorithm characteristics without having to perform a compute-expensive search.

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

# A Notes on repeatability and reproducibility

## A.1 Matching networks

Matching networks [36] first embed images into feature space with CNN, then adjusts and re-arranges the resulting vectors using bidirectional LSTM. Images from a support set are used for adjusting, and are coupled with the target image at query time. The query image class is chosen using $K$-nearest neighbours amongst the target image vector and the support image vectors. The authors were also the first to use the same scheme for training and testing, which improved the results of few shots learning.

**Official implementation** The authors did not provide an official implementation of their algorithm. However, [30] provided their reimplementation [29] which attains results comparable to the original article and is cited heavily by newer papers. This latest article also introduces proper new splits for miniImagenet, which have become a defacto standard for other experiments (see [27, 34])

**Challenges** The implementation of [29] is written in torch7. Although the authors provide a list of dependencies, the fact that torch7 is not actively supported, it provided additional challenges to run the code and reproduce results on a current operational environment. It took a lot of time and experimentation to find a combination of operation system, Cuda and dependency versions that works together and on current hardware: Cuda 8.0 is needed to run on the current hardware (Tesla P100, Tesla V100); Ubuntu 16.04 was needed instead of 14.04 to minimize manual installation and build from sources as much as possible (for Cuda 8.0 and libraries for integration); torch7, torch-autograd, torch-dataset installed from sources; some minor code changes were needed to work with newer torch-autograd versions; torch-ipc needed to be installed from particular git commit (as newer version introduced breaking changes), however C++ compiler flag for C++ standard needed to be changed to C++11 to be compatible with Cuda 8.0; "moses" library version <2.* (1.6.1.1) needed to be used as as 2.* version has different format for callbacks.

Several issues remained with the code: Cuda kernel recompilation is triggered on GPUs with compute capability 7.0 or higher when using Cuda 8, which required 16+Gb of memory for jobs and slowed startup time; though the code uses GPU, GPU utilization is low (less than 50%), while CPU utilisation remained high (3 to 4 cores are occupied during training); as different versions of libraries were chosen through experimentation, the code occasionally crashes with memory corruption, double resource de-allocation or Cuda drivers shutdown too early. We also found no way to set the code to run deterministically, as setting the random seeds of 'torch', 'lua math' and 'cutorch' did not eliminate randomness during training.

**Replicability effort** We ran the code with adaptations as described in A.1 **Challenges** inside a docker container on a kubernetes GPU cluster. We also made some code modifications to integrate it with our monitoring system, added early stopping and additional varying parameters like number of filters in CNN layers, LR decay rate and random seed. For the default parameters for matching networks without full conditional embedding (FCE) we got results better than those reported in Table 1 of the [30] : $53.57\%$ vs $51.09 \pm 0.71\%$. All further experiments have been run with FCE enabled, as this proved to be an improvement over basic embedding according to [36, 30].

## A.2 Prototypical Networks

In Prototypical Networks [34], the authors introduce an inductive bias in the form of prototype representations of each class. The model consists of a convolutional neural network learning a non-linear mapping of the input into an embedding space, in which a nearest neighbor classification can be performed by computing distances to those prototype representations. The classification relies on the squared Euclidean distance as a similarity measure, as the authors experimentally find that it outperforms the cosine distance for their settings. The article reports results on both the Omniglot and the miniImagenet datasets, for 5-shots and 1-shot experiments.

**Official implementation** The authors released their code in an official GitHub repository [33]. The available implementation only contains parameters and data loading functions for the Omniglot dataset and not miniImagenet. The repository is not currently maintained, with the last commit being in June 2018, with open issues dating back February 2018 un-addressed. The oldest and most commented-on issue asks the authors for a release of the detailed configuration [24] in a collective effort to reproduce the results.

Table 6: Top-3 accuracies over 600 test episodes for Prototypical Networks using Adam, using the default configuration (top), adding normalization (middle), and combining normalization with hyper-parameters search (bottom).

| Accuracy (IC 0.95) | Learning rate | LR decay period | Random seed |
|---|---|---|---|
| 59.01 ($\pm$ 0.73) | 0.001 | 20 | 8765 |
| 58.97 ($\pm$ 0.70) | 0.001 | 20 | 54321 |
| 58.93 ($\pm$ 0.69) | 0.001 | 20 | 5678 |
| 60.59 ($\pm$ 0.66) | 0.001 | 20 | 5678 |
| 59.41 ($\pm$ 0.61) | 0.001 | 20 | 7654 |
| 58.78 ($\pm$ 0.68) | 0.001 | 20 | 9876 |
| 62.50 ($\pm$ 0.53) | 0.005050 | 6 | 54321 |
| 62.21 ($\pm$ 0.54) | 0.003107 | 12 | 12345 |
| 62.03 ($\pm$ 0.54) | 0.005050 | 6 | 34567 |

**Replicability effort** We replicated the experimental conditions from the original article using a docker container. We used the original version for each technical component when specified (e.g. PyTorch 0.4, python 3.6) and the latest otherwise (e.g. cuda 9.1). Extending the released code base, we wrote a dataloader for miniImagenet using the data splits from [30], as mentioned in the original article. We used the set of hyper-parameters from the article when specified, and modified the implementation default set accordingly. We did set all random seeds as described in section 5.1. Some minor modifications were made to facilitate our large-scale analysis, such as passing the CUDA device as an argument or exporting the results to our monitoring system.

**Challenges** Despite the article reporting results on the miniImagenet dataset, the code released by the authors did not support that dataset. Considerable effort was needed to reproduce the experimental procedure. We did not expect this, as an official repository was made available for replicability purposes. Some training hyper-parameters were not specified in the article ; we found the missing values in other repositories reproducing the results [11, 27] and open issues discussions [24].

We were unable to reproduce the results from the article, obtaining 59.01% ($\pm$ 0.73) accuracy (see 6) in lieu of the expected 68.20% ($\pm$ 0.66) when running the default configuration:

- **Hyper-parameters** 64 filters in the hidden and output layers, 10,000 epochs, patience of 200 epochs, learning rate of $10^{-3}$, Adam optimizer, train/validation/test split from [30], PyTorch default BatchNorm2D epsilon of $10^{-5}$ and momentum of 0.1, learning rate linear decay gamma of 0.5 every 2,000 episodes
- **Training few-shot setting** 100 episodes, 20 ways, 5 shots, 15 query points
- **Testing few-shot setting** 600 episodes, 5 ways, 5 shots, 15 query points

Critically, while running the default configuration, we observed that the validation loss stopped improving during the first 20 epochs, i.e. 2,000 episodes. Therefore, the learning rate schedule appears to be ineffective and the patience of 200 epochs over-estimated.

In an effort to improve on these results from the default configuration, we normalized the input over miniImagenet (mean: 112.74, standard deviation: 68.72). Normalizing increased the accuracy to 60.59% ($\pm$ 0.66) (see Table 6).

We also varied multiple hyper-parameters values (see Table 3) and the optimizer, repeating the experiments using several random seeds. We obtained our best accuracy of 62.50% ($\pm$ 0.53) using Adam, with a learning rate of 0.005050 decaying every 6 epochs, a random seed of 54321, and all other hyper-parameters set to their default value.

