# OpenReview forum: "Reproducibility and Stability Analysis in Metric-Based Few-Shot Learning"
_ICLR.cc/2019/Workshop/RML — RML 2019_

### Official Review · AnonReviewer1 · 2019-04-04
**Interesting Paper but some parts difficult to understand**

**Rating:** 3
**Confidence:** 1

**Review:**

This paper studies reproducibility for few-shot learning.

My general impression after reading it is that it tries to cover a very large amount of ground and different number of subjects, which can make it a bit difficult to understand.  This paper doesn't just attempt to reproduce one paper, it attempts to reproduce three different papers and also proposes to use classical statistics tests with a mixed linear regression model to fit accuracies.  This breadth makes the paper's impact potentially very high, but also can make it difficult to understand and fully appreciate.

Some of the statistical tests could be given more thorough background information, as they may be outside of the range of knowledge of some readers in the ML community.

---

### Decision · Program_Chairs · 2019-04-05
**Acceptance Decision**

Accept